ecology

priority effects, desert rodents, competition, *Dipodomys*, generalized additive models, transient dynamics

**Author for correspondence:**
Erica M. Christensen
e-mail: erica.christensen@weecology.org

# Established rodent community delays recovery of dominant competitor following experimental disturbance

Erica M. Christensen[1,2], Gavin L. Simpson[3,4] and S. K. Morgan Ernest[1]

[1]Department of Wildlife Ecology and Conservation, University of Florida, 110 Newins-Ziegler Hall, Gainesville, FL 32611, USA
[2]New Mexico State University, Jornada Experimental Range, Wooton Hall, 2995 Knox Street, Las Cruces, NM 88003, USA
[3]Institute of Environmental Change and Society, and [4]Department of Biology, University of Regina, 3737 Wascana Parkway, Regina, Saskatchewan, Canada S4S 0A2

EMC, 0000-0002-5635-2502; GLS, 0000-0002-9084-8413; SKME, 0000-0002-6026-8530

Human activities alter processes that control local biodiversity, causing changes in the abundance and identity of species in ecosystems. However, restoring biodiversity to a previous state is rarely as simple as reintroducing lost species or restoring processes to their pre-disturbance state. Theory suggests that established species can impede shifts in species composition via a variety of mechanisms, including direct interference, pre-empting resources or habitat alteration. These mechanisms can create transitory dynamics that delay convergence to an expected end state. We use an experimental manipulation of a desert rodent community to examine differences in recolonization dynamics of a dominant competitor (kangaroo rats of the genus *Dipodomys*) when patches were already occupied by an existing rodent community relative to when patches were empty. Recovery of kangaroo rat populations was slow on plots with an established community, taking approximately 2 years, in contrast with rapid recovery on empty plots with no established residents (approx. three months). These results demonstrate that the presence of an established alternate community inhibits recolonization by new species, even those that should be dominant in the community. This has important implications for understanding how biodiversity may change in the future, and what processes may slow or prevent this change.

## 1. Introduction

Biodiversity in many ecosystems is changing in response to anthropogenic impacts [1–3], making it critical to understand the processes that accelerate change or impede our ability to reverse it. At the local scale, three main classes of interacting processes influence biodiversity: dispersal between patches [4], environmental conditions [5] and species interactions [6]. Alterations to any of these processes can alter local biodiversity. Dispersal links habitat patches, providing immigrants that can either rescue resident populations in danger of local extinction or introduce new species better suited to the local environment [4]. Changes in environmental conditions not only affect the physiological performance of resident species [7,8], but may also affect the outcome of competitive interactions [9] according to how well a particular habitat fulfils a species' niche requirements (e.g. abiotic tolerances, resource requirements). The network of species interactions (e.g. competitive interactions, predation pressures and disease susceptibilities) further restricts which species are found in which patches, even if all species can reach and survive in all the patches on a landscape. Shifts in these processes, especially shifts caused by human activities including reduced connectivity of patches (e.g. habitat fragmentation), landscape conversions (e.g. forest to

pastures) and exotic species introductions are blamed for much of the biodiversity change currently being observed [10–12].

Theoretically, restoring biodiversity should be as simple as restoring processes to their previous, pre-disturbance state. Ecological theory focused on stable states provides important insights into the conditions that allow communities to shift from one stable state to another (e.g. R* theory and coexistence theory, [13]). Nevertheless, the practice of restoring biodiversity to a particular state is often more difficult than we would expect, with unexpected outcomes or long delays in recovery. Increasingly, ecologists are discovering that in addition to studying stable states, we need to better understand transient dynamics—the dynamics that occur as a system moves from one state to another. Transient dynamics, while temporary, may persist for many generations of the target organism [14], move the system farther from the equilibrium state before approaching it [15], or increase the role of stochasticity in delaying or altering expected outcomes [16,17]. Thus, understanding the conditions that create prolonged transitory dynamics is a potentially critical component of understanding how and when biodiversity can be moved from one configuration to another.

Species interactions, often in combination with dispersal or environmental change, can play an important role in determining whether biodiversity shifts from one state to another as well as the dynamics of any resulting transient phase [16–18]. Species interactions can increase extinction probabilities [17] or make it difficult for new species to invade an existing community [19], even if resident species are inferior competitors to the invader (e.g. if inferior competitors have high initial abundances and display interference mechanisms) [20]. Species interactions can also influence the dynamics of how a community moves from one configuration to another through indirect mechanisms if those activities make the environment better for themselves and less suitable for other species (i.e. ecosystem engineers; [21]). Species interactions can delay or prevent expected state changes by slowing rates of change or causing increased instability during the transient phase [17]. Understanding the impact of species interactions on the temporal dynamics of communities as drivers shift them from one state to another is important, though often difficult to study, especially for multi-species assemblages under field conditions.

We examined the role of species interactions in impeding biodiversity change using a replicated long-term experiment that manipulated the dispersal of seed-eating rodents into experimental plots (figure 1). The three levels of our manipulation (rodent removals, kangaroo rat removals and controls) created a landscape with plots containing different rodent communities ranging from undisturbed (controls), to lacking a single genus of behaviourally dominant seed eaters (kangaroo rats of genus *Dipodomys*), to consisting of only a few transient individuals (rodent removals). Many of our rodent species are territorial and sequester resources in caches, providing exactly the scenario where inferior competitors may be expected to delay colonization of a dominant species. Additionally, because kangaroo rats are important granivores in this system, their removal affects the annual plant community that serves as their resource base [22,23]. In 2015, we converted half of our kangaroo rat and rodent removal plots to controls, allowing rodents to recolonize. We compared the recovery on the newly opened rodent removal and kangaroo rat removal treatments to our unchanged

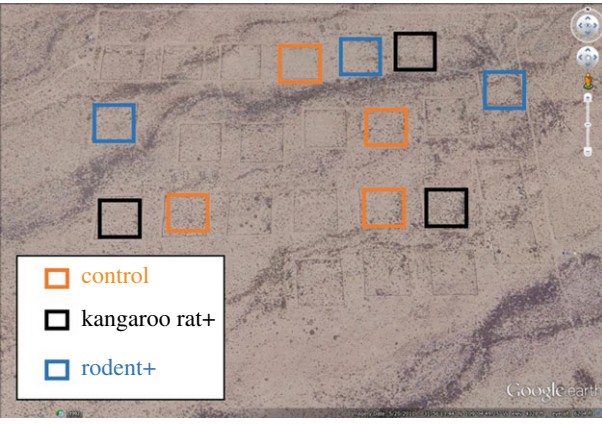

**Figure 1.** Map of the study area showing 10 experimental plots used for these analyses. The outlines of 14 additional plots not used for these analyses, but which were subjected to similar experimental manipulations, can be faintly seen. Map data from Google earth. (Online version in colour.)

long-term control plots using generalized additive models (GAMs) to assess whether there were differences in the dynamics of the re-invading kangaroo rats on these different types of plots. We also examined the dynamics of the other seed-eating rodents and metabolic flux of the entire rodent community to assess the roles of direct and indirect species interactions in explaining the recolonization dynamics of kangaroo rats. Finally, we examined differences in plant species composition among the treatments to assess the impact of differences in the plant community caused by manipulating the rodent community.

## 2. Site and methods

The experiment was conducted at the Portal Project, a 20 ha study site, 6 km northeast of the town Portal, Arizona located on unceded land of the Chiricahua Apache in the Chihuahuan Desert [24]. Twenty-four 50 m by 50 m experimental plots are enclosed by a 50 cm high fence, and rodent access is controlled by gates in the fences at ground level. The size and/or absence of gates regulates access to the plot: large gates for controls, smaller gates for kangaroo rat removals and no gates for rodent removals. Each plot was trapped every month using 49 Sherman traps, and information from captured rodents was recorded including species, sex and weight. Any rodents caught on total removal plots, and kangaroo rats caught on kangaroo rat removal plots, were recorded and relocated at least a quarter mile away from the site. Because there are two relatively distinct periods of annual plant growth with little overlap in species, plant abundances were sampled on each plot twice each year—once to capture the summer community and again to capture the winter community. On each plot, all plants were identified and counted on sixteen 0.25 m$^2$ quadrats placed at permanently marked locations.

While data collection began at this site in 1977, this analysis focuses on the time period 2013–2018. In March 2015, we changed the treatments on 12 of the 24 plots. The pre-existing treatments on those 12 altered plots had been in place continuously since 1989. We focus here on three specific treatments: (i) kangaroo rat+ plots: kangaroo rat removal plots that had gates enlarged to become controls (three plots), which allowed the addition of kangaroo rats to pre-existing rodent communities; (ii) rodent+ plots: rodent removal plots that had new

gates created to become controls (three plots), which allowed the addition of all rodent species to those plots; and (iii) controls: plots that were control plots for the entire period 2013–2018 (four plots), which serve as our reference plots for assessing the dynamics of our newly created control plots. All data and code for these analyses are available on GitHub (https:// github.com/emchristensen/PlotSwitch) and archived on Zenodo [25].

## (a) Time-series analysis using generalized additive models

We used GAMs to assess the effect of treatment on various rodent community metrics over time, using the R package mgcv v. 1.8–23 [26] for R 3.5.1 [27]. We constructed a GAM for each metric including separate smooths of time for each pre-2015 treatment type, then computed the difference between pairs of treatment-specific smooths, focusing on comparing kangaroo rat+ plots to long-term controls, and rodent+ plots to long-term controls. Where the Bayesian credible intervals for the difference in smooths included zero, we interpreted this to mean there was no effective difference in the metric of interest for that pair of smooths.

To assess the ability of competitively dominant kangaroo rats to colonize suitable patches, we constructed time series of pooled number of individuals of the three species of kangaroo rats found at the site (*Dipodomys merriami*, *Dipodomys ordii* and *Dipodomys spectabilis*) on each plot over time. Because we expect inferior competitors to be displaced by the invasion of kangaroo rats, we also constructed time series of pooled numbers of individuals of non-kangaroo rat seed-eating species (10 species from five genera: *Baiomys taylori*, *Chaetodipus baileyi*, *Chaetodipus penicillatus*, *Perognathus flavus*, *Peromyscus eremicus*, *Peromyscus leucopus*, *Peromyscus maniculatus*, *Reithrodontomys fulvescens*, *Reithrodontomys megalotis* and *Reithrodontomys montanus*). Because these two time series consist of count data, we used a Poisson distribution in both GAMs. We found that there were some small but significant differences between plots within treatments, and so included plot-specific smooths in the GAMs as well.

To examine effects at the community level, we analysed time series of community metabolic flux of the seed-eating rodent community on each plot. Total metabolic flux represents an estimate of community size and resource uptake of the community as a whole [28] and is generally less variable through time than species composition or species abundances [29]. Metabolic rate of an individual organism scales with body size according to the equation: $E \propto m^{(3/4)}$ where $E$ is the metabolic rate (or energy) and $m$ is the mass of the individual [30]. We estimated metabolic flux of individual captured rodents based on measured masses and summed by plot and time step to obtain total community metabolic flux. We fitted a GAM to the resulting time series, using a Tweedie distribution, and plot-specific smooths as well.

## (b) Plant community analysis

Differences in plant community composition were assessed using a partial canonical correspondence analysis (pCCA) and permutational significance tests, controlling for between-year effects (using R package vegan, v. 2.5-2 [31]). We square root transformed the plant abundance data to account for large differences in total abundance between years and species.

Owing to project funding gaps, plant data were not collected in all years leading up to the treatment change in 2015. We used all data available going back to 2008, which amounted to 3 years of data for the summer annual community (2008, 2011 and 2014) and 5 years of data for the winter annual community (2008, 2012, 2013, 2014 and 2015).

## (c) Population analyses

To better understand the observed differences in population size of kangaroo rats between treatments, we performed population-level analyses on these species during the period after the treatment change. We examined the rate that new kangaroo rat individuals entered each plot type, and performed a multistrata population model using the package RMark (v. 2.2.6; [32]), a package which uses the MARK software [33] to estimate survivorship (*S*) of kangaroo rat species on each experimental treatment type and probability of transition (Psi) between treatment types. We used Akaike information criterion to select the best model from candidate models with or without effect of strata (in this case, strata represents experimental treatment type) on *S* and Psi. Analysis was performed on *D. merriami* and *D. ordii;* there were too few captured individuals of *D. spectabilis* for modelling.

## 3. Results and discussion

The presence of an existing rodent community had a substantive impact on dynamics as the plots transitioned to the expected control state. Recovery of kangaroo rat populations occurred more slowly on plots containing a pre-existing rodent community than on plots where rodents were not already present (figure 2a). Differences between the treatment-specific smooths from the GAM (figure 2b) showed that kangaroo rat abundances on rodent+ plots converged to control levels within three months (i.e. the 95% credible interval on the difference between the control and rodent+ models overlapped 0 starting in June 2015). While kangaroo rat populations on rodent+ plots converged with controls quickly, they also continued to experience large population oscillations that sometimes caused abundances to exceed those on long-term controls for brief periods of time, even after the treatment types converged on average. This is consistent with theoretical work demonstrating that transient dynamics sometimes amplify the initial perturbation, moving the system farther from the equilibrium point before converging to it [15,34]. By contrast, kangaroo rat+ plots—which had a pre-existing rodent community—required an additional 21 months for kangaroo rat populations to converge to control levels (figure 2b). Unlike rodent+ plots, kangaroo rat populations on the kangaroo rat+ plots appeared less oscillatory and never exceeded abundances on control plots (figure 2b). Thus, the treatment types differ in their transient dynamics during the colonization and population increase in kangaroo rats. Because all treatments are embedded in the same habitat matrix where kangaroo rats are an abundant species (figure 1), all plots have an equal number and size of gates in fences, and treatments are interspersed across the site (figure 1), the differences in the transient dynamics between rodent+ and kangaroo rat+ plots appear to be related to the presence or absence of a pre-existing rodent community.

Differences between treatments in the influx (birth, immigration) and efflux (death, emigration) of individuals drove the observed contrasts in transient dynamics on rodent+ and

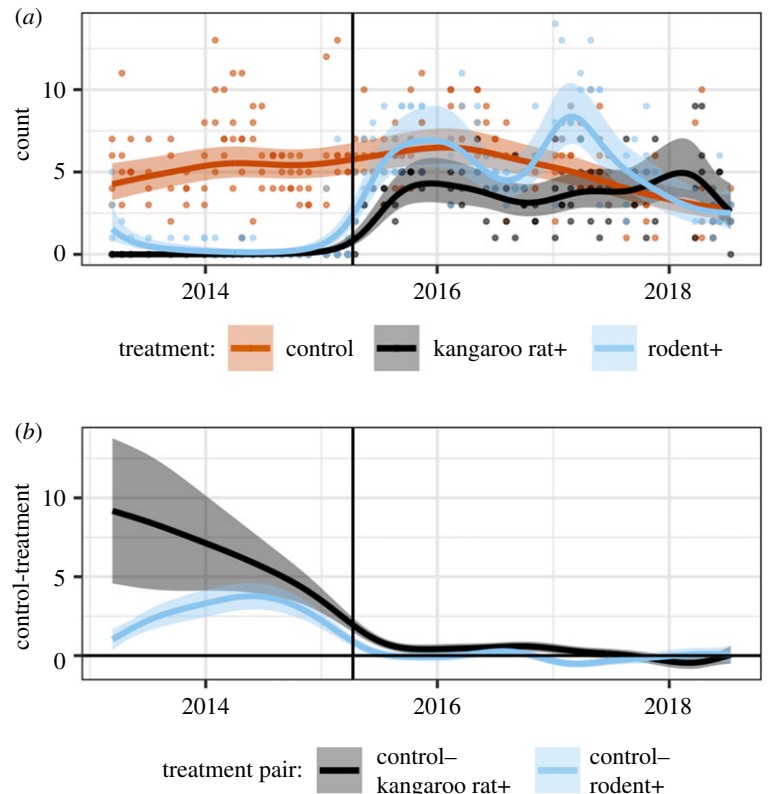

**Figure 2.** (*a*) GAM model of number of kangaroo rats per plot and (*b*) difference of treatment-specific smooths from GAM, shown on the link (log) scale. Vertical lines mark when treatments were changed in March 2015. (Online version in colour.)

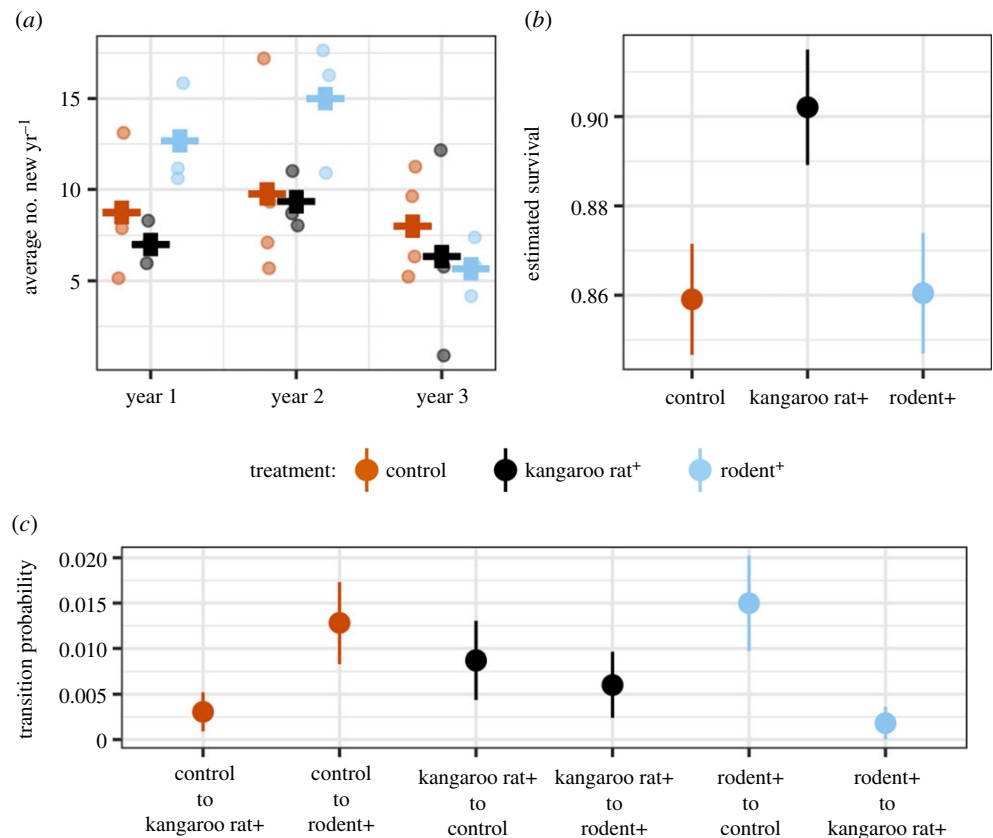

**Figure 3.** Results of population models for *D. merriami*. (*a*) Average number of new individuals by treatment and year since treatment change. Points represent total new individuals on a plot; horizontal bars represent average by treatment; (*b*) survivorship estimate by treatment type; (*c*) estimated probability of an individual moving between treatment types. (Online version in colour.)

kangaroo rat+ plots in complicated ways, which differ based on which kangaroo rat species is examined. When rodent removal plots were converted to controls, they received substantively higher numbers of new kangaroo rat individuals on average than either long-term control plots or kangaroo rat+ plots (figure 3*a*, see also the electronic supplementary material,

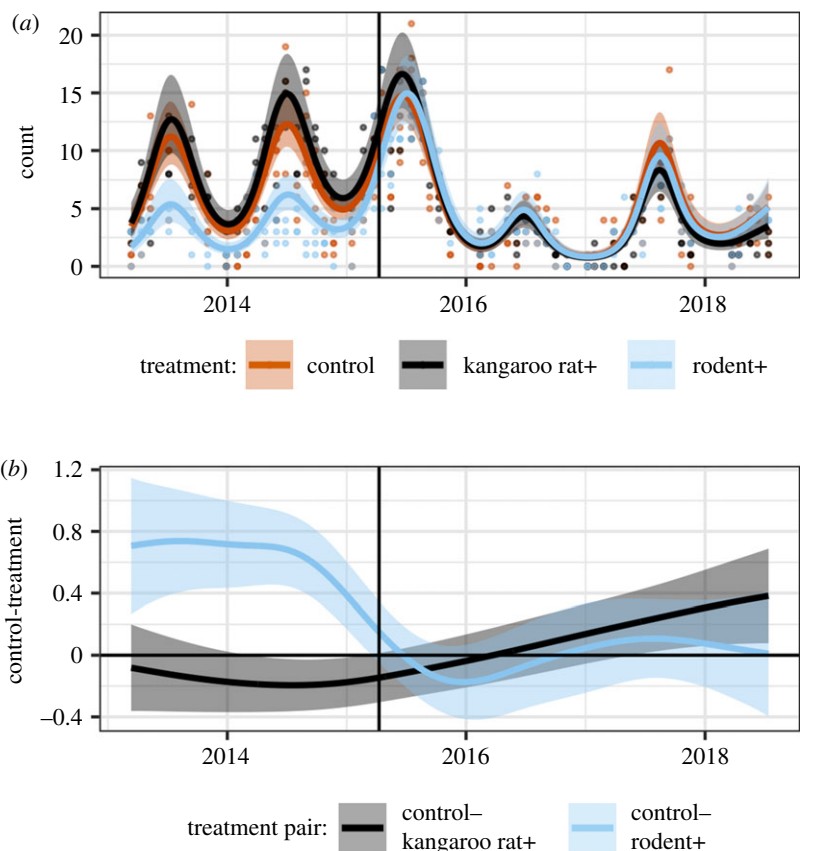

**Figure 4.** (a) GAM model of abundances of non-kangaroo rat species per plot and (b) difference of treatment- specific smooths from GAM, shown on the link (log) scale. Vertical lines mark when treatments were changed in March 2015. (Online version in colour.)

appendix tables S1 and S2). This high influx of new individuals lasted 2 years despite the nearly immediate convergence with control plots in terms of overall kangaroo rat abundances, and was driven primarily by *D. merriami,* the more abundant kangaroo rat species at the site. While new individuals flooded into rodent+ plots, kangaroo rat+ plots exhibited slightly suppressed numbers of new arrivals compared to control plots, despite the fact that these plots—like the rodent+ plots—also initially lacked kangaroo rats. Transition probabilities suggest that the suppressed influx of *D. merriami* individuals was in part owing to immigration. *Dipodomys merriami* individuals captured on rodent+ or control plots were much less likely to move to a kangaroo rat+ plot than to another treatment (figure 3c; *D. ordii* showed no treatment effect; electronic supplementary material, tables S3 and S4). While the transition probabilities and numbers of new individuals suggest that some mechanism reduced the influx of *D. merriami* individuals onto kangaroo rat+ plots, once a *D. merriami* individual settled on a kangaroo rat+ plot, it experienced a higher survivorship and extremely low probability of moving to another treatment type (figure 3b,c). By contrast, *D. ordii* exhibited little difference in survival on rodent+ versus kangaroo rat+ plots, though its survival estimate was lower on those treatments than on controls (electronic supplementary material, table S4). This complicated picture suggests that the difference in transient dynamics between treatments is owing to some condition on kangaroo rat+ plots that slightly suppressed the influx of new individuals, potentially delaying recovery on kangaroo rat+ plots, despite the fact that conditions on kangaroo rat+ plots did not negatively impact survival or generate increased movement to another treatment type.

Because plots at the site experience the same weather events and do not differ in their distance from source populations, it

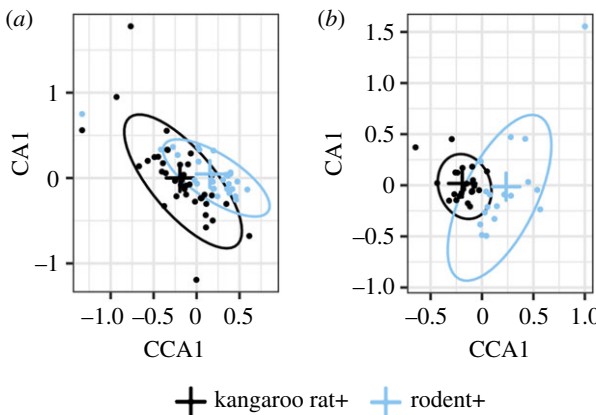

**Figure 5.** pCCA of treatment effects on plant composition for (a) winter and (b) summer annual plants. Crosses indicate data centroids by treatment, and ellipses enclose 95% of the data points. (Online version in colour.)

seems logical that the difference in kangaroo rat transient dynamics is related to whether or not a plot had a pre-existing rodent community at the time the plot was converted to a control. A pre-existing rodent community could potentially impact the colonization dynamics of kangaroo rats either directly through competitive interactions (e.g. territoriality) or indirectly through impacts on seed availability or the plant community. Examination of the dynamics of the non-kangaroo rat species (i.e. small granivores) suggests that competitive interactions are an unlikely explanation for the differences in transient dynamics. Before the change in treatments in 2015, abundances of the non-kangaroo rat rodents on kangaroo rat removals were higher than on controls plots (kangaroo rat removals averaged 7.8 individuals plot$^{-1}$ month$^{-1}$; controls

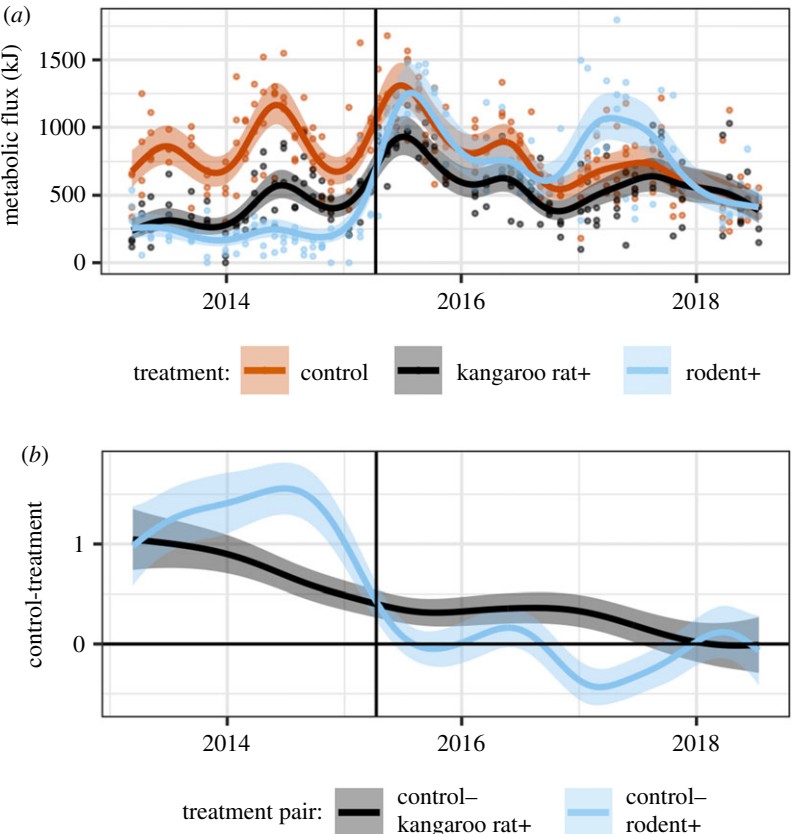

**Figure 6.** (a) GAM model of community-level metabolic flux per plot and (b) difference between treatment-specific smooths from GAM, shown on the link (log) scale. Vertical lines mark when treatments were changed in March 2015. (Online version in colour.)

5.7 individuals) and much lower than control levels on rodent removals (3.1 individuals plot$^{-1}$ month$^{-1}$) (see the electronic supplementary material, appendix: figure S1 and table S5). After all plots were converted to controls, non-kangaroo rat abundances on both the rodent+ and kangaroo rat+ plots quickly converged to control levels within a few months (figure 4b). The rapid decline in non-kangaroo rat species is consistent with previous research showing that kangaroo rats are behaviourally dominant over the other, typically smaller, seed-eating rodents [35,36] (see also the electronic supplementary material, appendix figure S2). Because differences between treatments in non-kangaroo rat species abundances disappeared quickly, it seems unlikely that direct interference by the non-kangaroo rat species explains the 21-month delay in the recovery of kangaroo rats on kangaroo rat+ plots.

While direct effects of rodent interactions seem an unlikely cause for the different transient dynamics on kangaroo rat+ plots, the pre-existing community could have indirectly impacted kangaroo rat colonization through their pre-2015 impacts on the plant community. Rodents can impact the plant community as consumers, and the plant community can also impact the rodent community through two mechanisms: habitat structure [37] and the resource base that vegetation provides [38]. If the plant community was a factor impeding colonization by kangaroo rats, we would expect to see strong pre-2015 differences in plant composition between the kangaroo rat removal plots and rodent removal plots. We conducted a pCCA by treatment for the 7 years leading up to the treatment change in 2015 (figure 5) to assess differences in the plant community while controlling for between-year variation. The effect of treatment was significant for both the winter plant community (pCCA permutation test: $R^2 = 0.03$ and $p = 0.002$) and the

summer plant community (pCCA permutation test: $R^2 = 0.04$ and $p = 0.004$). In both cases, however, the proportion of variance explained by treatment was small (less than 5%). Thus, while the experimental manipulations did impact plant composition, these impacts were only weakly different between the rodent removals and kangaroo rat removals during this time period (figure 5; see the electronic supplementary material, appendix figures S3 and S4). Our results do not rule this out as a possible mechanism, but support for this as a potential driver of differences in transient dynamics is currently weak.

While there is only weak support for the resident community impacting the transient dynamics on kangaroo rat+ plots through an effect on the plant community, desert rodents in this system can also impact resource availability by sequestering resources. One option for why kangaroo rats arrived in greater numbers on rodent+ plots than kangaroo rat+ plots is that resources—in this case seeds—on rodent+ plots were more readily available to new colonizers. Seeds on rodent+ plots may have been more plentiful owing to lack of consumption by an established rodent community. These seeds may also have been easier to find because they had not been gathered and deposited into caches [36,39]. While we do not have information about the distribution or availability of seeds, we can gain some insights via examination of metabolic energy flux of the rodent community on the different treatments. Community metabolic flux [28,30] is a measure of the energy intake rate required to sustain the rodent community on a plot, and thus is an index of resource use by the community as a whole. If there were more resources available on rodent+ plots because of seed accumulation, then we would expect the metabolic energy flux on these plots to be higher than on controls. However, there is little evidence that this

was the case (figure 6) except for a brief period of time beginning in 2017, which was over a year after the treatments were changed. By contrast, metabolic flux on kangaroo rat+ plots remained below both rodent+ and control plots until 2018. The decline in the non-kangaroo rat rodents early in the recovery process (figure 4) should have freed up resources for the kangaroo rats to use; however, our results suggest that there were still unused resources available on the kangaroo rat+ plots that the kangaroo rats were not accessing. The fate of these missing resources is unknown. It is possible that these resources were hidden in caches that kangaroo rats were simply unable to find or that those resources were pre-empted by one of the other seed-eating taxa in this ecosystem (i.e. ants or birds). Unfortunately, we are currently unable to test either of these hypotheses.

## 4. Conclusion

While we cannot explain the mechanism by which a resident community impacts the transient dynamics as a community moves from one state to another, our results clearly show that lack of a resident community resulted in rapid convergence but more variable dynamics, while the presence of a resident community created a longer transient period before convergence. The scale and nature of our experiment suggest that the ability of a resident community to impact the population dynamics of a dominant competitor can be unexpectedly strong and have major implications for the length of the transient period. The large source pool of immigrants from the habitat matrix should have provided a strong invasion advantage to the kangaroo rats, and the relatively small plot size (50 m × 50 m) ensured that only dispersal (not population growth) was required for kangaroo rats to establish control-level populations on experimental plots. Despite these advantages, we observed transient dynamics in kangaroo rat recolonization of experimental plots. The most likely mechanism for this delay is that the previously established species displayed interference competition by sequestering resources in caches, with potentially some role of vegetation change, making these plots less appealing to kangaroo rat immigrants for approximately 2 years. In our case, transient dynamics eventually led to restoration of the control state; however, prolonged periods of transient dynamics provide opportunity for unexpected and complex dynamics, especially in the presence of stochasticity, a common feature of ecological systems [14,40].

Many common methods of managing wildlife populations (e.g. opening/closing dispersal corridors, translocating individuals) will tend to produce transient dynamics [41]. In our experiment, we showed that restoring the dispersal ability of kangaroo rats resulted in unexpected transient dynamics owing to the complicating effects of species already present in areas targeted for colonization. Although our experimental plots were indistinguishable from control plots in the long term, the short-term dynamics revealed valuable information about the processes operating in the system. These results suggest not only that previously established species can alter or delay return to the desired biodiversity state, but also the reverse—that removal or disruption of an existing community may facilitate and/or speed up biodiversity shifts. This result is consistent with other work that has noted rapid shifts in biodiversity states after disturbances [42–44]. Thus, disturbances that eliminate the impacts or advantages of an existing community may create conditions that make biodiversity state shifts more probable. Understanding short-term dynamics is critically important to applied ecology. Because the length of conservation projects tends to be finite and short, focusing solely on asymptotic stable-state dynamics may lead to misleading results that affect management decisions [45]. While our study suggests that the presence of a resident community can alter and lengthen the transient phase, further work—including replicating experiments such as these under different conditions—are required to better understand how differences in the resident community or other initial conditions may generate and modify transient dynamics.

Ethics. Animal research was reviewed and approved by the University of Florida Institutional Animal Care and Use Committee (IACUC protocol no. 201808839) and followed the guidelines of the American Society of Mammalogists for use of wildlife in research. This study was also conducted under a state of Arizona Scientific Collecting Licence (most recently SP643824).

Data accessibility. Data and R code used for these analyses are available from Zenodo: https://doi.org/10.5281/zenodo.3460769 [25].

Competing interests. We declare we have no competing interests.

Funding. This work was supported by grant no. DEB-1622425 from the National Science Foundation. E.M.C. was also supported by funds to the USDA-ARS Jornada Experimental Range no. 3050-11210-009-00-D. G.L.S. was supported by an NSERC Discovery Grant (grant no. 2014-04032).

Acknowledgements. We would like to thank D. Valle, T. Palmer and B. Baiser for their helpful comments on the manuscript, and the anonymous reviewers and editors for their suggestions for improvement. We would also like to thank E. Bledsoe for advice and code used to perform the population models.

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
