## [Reviewer comments · Proceedings of the Royal Society B: Biological Sciences]

Review History

RSPB-2019-0188.R0 (Original submission)

Review form: Reviewer 1

Recommendation

Reject – article is scientifically unsound

Scientific importance: Is the manuscript an original and important contribution to its field?

Acceptable

General interest: Is the paper of sufficient general interest?

Good

Quality of the paper: Is the overall quality of the paper suitable?

Good

Is the length of the paper justified?

Yes

Should the paper be seen by a specialist statistical reviewer?

No

Do you have any concerns about statistical analyses in this paper? If so, please specify them explicitly in your report.

No

It is a condition of publication that authors make their supporting data, code and materials available - either as supplementary material or hosted in an external repository. Please rate, if applicable, the supporting data on the following criteria.

Is it accessible?

N/A

Is it clear?

N/A

Is it adequate?

N/A

Do you have any ethical concerns with this paper?

No

Comments to the Author

The authors presented a multi-yr manipulated exexperiment on recovery of rodent communities. The results showed established species might impede the restoration of kangaroo rats. I appreciate the effort of such an experiment. However, the conclusion is not surprising. I would expect some results on the mechanism of slow recovery for the kangaroo removal treatment, but only the plant community was compared. For example, the demography, habitat use, behavior, dispersal process? Essentially, the authors did not answer why on earth the recovery or resembling-control-group is slower. Much more content is needed. If they can provide more information, this should be an outstanding paper.

1. The authors used count and metabolic flux to test restoration of kangaroo rats. The result is simple and straitforward, with many things unexplored. A deeper digging is thus necessary. Eg., for a community, the descriptions of the state are many, such as some common ones like diversity indices, etc. at population level, reproduction, age structure, sex ratio, etc. Pls do not misunderstand that I am asking for a full list of properties. I just suggest more exploration on what was exactly recovered.
2. The control declined in the last two years. Any idea why? Except just saying a system phenomena.
3. I am not sure, but is there any possibility the rodents move between control and treatments. If yes, how does it affect the results?
4. In fig. 2, the seasonal pattern is missing compared to non-kangaroo rodents in control group. is gam smooth not appropriate (need more knots in smooth function?) or actually missing?

Review form: Reviewer 2

Recommendation

Accept with minor revision (please list in comments)

Scientific importance: Is the manuscript an original and important contribution to its field?

Good

General interest: Is the paper of sufficient general interest?

Good

Quality of the paper: Is the overall quality of the paper suitable?

Good

Is the length of the paper justified?

Yes

Should the paper be seen by a specialist statistical reviewer?

Yes

Do you have any concerns about statistical analyses in this paper? If so, please specify them explicitly in your report.

No

It is a condition of publication that authors make their supporting data, code and materials available - either as supplementary material or hosted in an external repository. Please rate, if applicable, the supporting data on the following criteria.

Is it accessible?

Yes

Is it clear?

Yes

Is it adequate?

Yes

Do you have any ethical concerns with this paper?

No

Comments to the Author

Established rodent community delays recovery of dominant competitor following experimental disturbance by Christensen et al.

Proceedings of the Royal Society B: manuscript #RSPB-2019-0188

This study looked at the length of time for three species of kangaroo rats (*Dipodomys* spp.) to establish on 50 m² plots that either had an intact rodent community without kangaroo rats or on plots that had been kept free of all rodents. The authors used control plots with all species as a comparison. The work took place in desert habitat near Portal in southeastern Arizona. Multiple papers have been written from the experimental design at this site since the late 1970s. The basic outcome was that kangaroo rats established on plots without any rodents in about 3 mo, but it took about 2 y for a population of kangaroo rats to establish on plots with and otherwise intact rodent community.

The methodology is sound and the manuscript is well written. I have no major methodological or interpretive concerns. I do, however, have two minor comments and several small edits. Although this study has been internally replicated, this study needs to be repeated in time at least

once to determine if the same delayed recovery of kangaroo rats on established rodent communities gives the same outcome as this one-time study. I am not suggesting that this study is fatally flawed, and although they allude to this problem in the Discussion, I think you need to more forcefully acknowledge that outcomes may be different at different times. Also, you point out that what you studied bears on restoration ecology, but you do not come back to this topic in the Discussion. I think this is an important topic, and I note that the time differences for kangaroo rats you found actually is unimportant to reestablishing *Dipodomys*. In fact, this is encouraging in that kangaroo rats will reestablish rather quickly even with multiple competitors at a site.

Minor edits:

Lines 70, 99, 115, others: You use 'which' in these instances when the correct word is 'that.' That is used for restrictive clauses, and which is for unrestrictive clauses and is set off by a comma.

Line 110 and others: Remove the hyphen from behaviorally-dominant and other ly words. Words that end in ly are not hyphenated when paired with another adjective.

Line 126: 20-hectare change to 20-ha. Throughout the manuscript, use standard abbreviations for units.

Line 127: Capitalize desert in Chihuahuan Desert.

Line 133: Use only metric system throughout the paper.

Line 159: Change since to because. Since is only used when referencing time periods, and because is used for causality.

Throughout the manuscript, you use However to start sentences. Because however is a conjunctive adverb, it should lower case, and set off from the preceding sentence with a semicolon and followed by a comma (...; however,...).

Decision letter (RSPB-2019-0188.R0)

28-Mar-2019

Dear Dr Christensen:

I am writing to inform you that your manuscript RSPB-2019-0188 entitled "Established rodent community delays recovery of dominant competitor following experimental disturbance" has, in its current form, been rejected for publication in Proceedings B.

This action has been taken on the advice of referees, who have recommended that substantial revisions and further work are necessary. With this in mind we would be willing to consider a resubmission, provided the comments of the referees are fully addressed. It is important to note that this is not a provisional acceptance, your resubmission will need to fully take into account the different issues and concerns that are in the reports at the end of this email.

The resubmission will be treated as a new manuscript. However, we will approach the same reviewers if they are available and it is deemed appropriate to do so by the Editor. Please note

that resubmissions must be submitted within six months of the date of this email. In exceptional circumstances, extensions may be possible if agreed with the Editorial Office. Manuscripts submitted after this date will be automatically rejected.

Please find below the reports submitted by the referees, not including confidential comments and issues submitted to the Editor.

If you do choose to resubmit your manuscript, please upload the following:

Sincerely,

Proceedings B
 mailto: proceedingsb@royalsociety.org

=====

Associate Editor
 Comments to Author:

Any field study that has been going on as long as the Portal site study with mostly-consistent methods and plots, is valuable! I am glad that a new cohort of researchers has taken it over and is keeping it going. And the results you present are intriguing.

However, the results are oddly incongruent with the focus of large parts of the Introduction and Conclusion. There is no evidence of alternative stable states, so why focus on this issue in the Introduction and Conclusion? I am just guessing - but perhaps you focused on alternative stable states in your NSF proposal? Whether or not this was the case, the Introduction and Conclusion of the paper should fit the actual results. Specifically, the Introduction, from line 77 through the following paragraph, is written as if you were providing the frame for a finding of alternative stable states. Lines 306 - 314 in the Conclusion are also about alternative stable states. I recommend rethinking and completely focusing the Introduction and Conclusion so they provide a frame for what you actually found: differences in the speed (and dynamics, see below) of recovery, with no evidence of a second stable state.

You have some good text about the transient dynamics and the relative speed of recovery, which could make a better focus than alternative stable states. In line 326, you state that "Our results suggest not only that existing established species can alter or delay changes in biodiversity state, but also the reverse - that removal or disruption of an existing community [rodent removal treatment, I gather] may facilitate and/or speed up biodiversity shifts [kangaroo recovery, I gather]." I found this very interesting, and I think other readers would too. It might also provide a good bridge to discussing the restoration implications, which, as Reviewer 2 points out, you mention in the Introduction but do not come back to in the Discussion or Conclusion. Reviewer 2

considers that the difference in recovery times between the two treatments is actually too small to be important - this should be addressed if you discuss the results in the context of restoration.

Reviewer 1 notes that in Fig 2B and 3B, almost all traces of the seasonal oscillations are gone. I concur with this reviewer's concern that information may be being lost due to smoothing. The rodent removal had much stronger oscillations than the other two (Fig 2A). And most of the time, there were more kangaroo rats in the rodent removal (blue) than in the kangaroo rat removal (gray) and sometime more than in the control (orange). You say (line 191 and ff) that kangaroo rat numbers in rodent removals (blue) converged with control numbers (orange) within 3 months - which they did, but then they bounced around, including an overshoot. Why did the rodent removal plots have so much greater oscillations in kangaroo rat numbers than the controls or rodent removals? Did this have anything to do with faster recovery there?

As Reviewer 1 says, this paper leaves us with a puzzle: what caused the difference in the rate of kangaroo rat recovery between the two removal treatments? You eliminate differences in overall plant community composition and total resource availability. Reviewer 1 suggests some other possibilities. What do you think is the most likely explanation? I concur with this reviewer: "If they can provide more information, this should be an outstanding paper."

===

Reviewers' Comments to Author:

Referee: 1

The authors presented a multi-yr manipulated exexperiment on recovery of rodent communities. The results showed established species might impede the restoration of kangaroo rats. I appreciate the effort of such an experiment. However, the conclusion is not surprising. I would expect some results on the mechanism of slow recovery for the kangaroo removal treatment, but only the plant community was compared. For example, the demography, habitat use, behavior, dispersal process? Essentially, the authors did not answer why on earth the recovery or resembling-control-group is slower. Much more content is needed. If they can provide more information, this should be an outstanding paper.

1. The authors used count and metabolic flux to test restoration of kangaroo rats. The result is simple and straightforward, with many things unexplored. A deeper digging is thus necessary. Eg., for a community, the descriptions of the state are many, such as some common ones like diversity indices, etc. at population level, reproduction, age structure, sex ratio, etc. Pls do not misunderstand that I am asking for a full list of properties. I just suggest more exploration on what was exactly recovered.

2. The control declined in the last two years. Any idea why? Except just saying a system phenomena.

3. I am not sure, but is there any possibility the rodents move between control and treatments. If yes, how does it affect the results?

4. In fig. 2, the seasonal pattern is missing compared to non-kangaroo rodents in control group. is gam smooth not appropriate (need more knots in smooth function?) or actually missing?

===

Referee: 2

Established rodent community delays recovery of dominant competitor following experimental disturbance by Christensen et al.

Proceedings of the Royal Society B: manuscript #RSPB-2019-0188

This study looked at the length of time for three species of kangaroo rats (*Dipodomys* spp.) to establish on 50 m² plots that either had an intact rodent community without kangaroo rats or on plots that had been kept free of all rodents. The authors used control plots with all species as a comparison. The work took place in desert habitat near Portal in southeastern Arizona. Multiple papers have been written from the experimental design at this site since the late 1970s. The basic outcome was that kangaroo rats established on plots without any rodents in about 3 mo, but it took about 2 y for a population of kangaroo rats to establish on plots with and otherwise intact rodent community.

The methodology is sound and the manuscript is well written. I have no major methodological or interpretive concerns. I do, however, have two minor comments and several small edits. Although this study has been internally replicated, this study needs to be repeated in time at least once to determine if the same delayed recovery of kangaroo rats on established rodent communities gives the same outcome as this one-time study. I am not suggesting that this study is fatally flawed, and although they allude to this problem in the Discussion, I think you need to more forcefully acknowledge that outcomes may be different at different times. Also, you point out that what you studied bears on restoration ecology, but you do not come back to this topic in the Discussion. I think this is an important topic, and I note that the time differences for kangaroo rats you found actually is unimportant to reestablishing *Dipodomys*. In fact, this is encouraging in that kangaroo rats will reestablish rather quickly even with multiple competitors at a site.

Minor edits:

Lines 70, 99, 115, others: You use 'which' in these instances when the correct word is 'that.' That is used for restrictive clauses, and which is for unrestrictive clauses and is set off by a comma.

Line 110 and others: Remove the hyphen from behaviorally-dominant and other ly words. Words that end in ly are not hyphenated when paired with another adjective.

Line 126: 20-hectare change to 20-ha. Throughout the manuscript, use standard abbreviations for units.

Line 127: Capitalize desert in Chihuahuan Desert.

Line 133: Use only metric system throughout the paper.

Line 159: Change since to because. Since is only used when referencing time periods, and because is used for causality.

Throughout the manuscript, you use However to start sentences. Because however is a conjunctive adverb, it should be lower case, and set off from the preceding sentence with a semicolon and followed by a comma (...; however,...).

Author's Response to Decision Letter for (RSPB-2019-0188.R0)

See Appendix A.

RSPB-2019-2269.R0

Review form: Reviewer 2

Recommendation

Accept with minor revision (please list in comments)

Scientific importance: Is the manuscript an original and important contribution to its field?

Good

General interest: Is the paper of sufficient general interest?

Good

Quality of the paper: Is the overall quality of the paper suitable?

Good

Is the length of the paper justified?

Yes

Should the paper be seen by a specialist statistical reviewer?

No

Do you have any concerns about statistical analyses in this paper? If so, please specify them explicitly in your report.

No

It is a condition of publication that authors make their supporting data, code and materials available - either as supplementary material or hosted in an external repository. Please rate, if applicable, the supporting data on the following criteria.

Is it accessible?

Yes

Is it clear?

N/A

Is it adequate?

N/A

Do you have any ethical concerns with this paper?

No

Comments to the Author

Established rodent community delays recovery of dominant competitor following experimental disturbance by Christensen et al.

Proceedings of the Royal Society B: manuscript #RSPB-2019-2669

This is a revision of the paper I reviewed in the Spring. I did not have many serious criticisms of the original paper. I pointed out that the study needed to be repeated in time at least once to determine if the same delayed recovery of kangaroo rats on established rodent communities gives the same outcome as this one-time study and suggested at least pointing this out in the

Discussion. The authors have done this. I also wanted to see some discussion of restoration ecology in the Discussion based on these findings, and this was done, although I think the topic could have been better and more fully explained. That said, what the authors have written is fine.

There still are minor edits to make:

Lines 211: Delete *In order* from the beginning of the sentence. It is clear if you write "To better...." Remove *In order* anywhere it occurs.

Lines 241, 304, etc.: Change *since* to *because*. *Since* is only used when referencing time periods, and *because* is used for causality. You did this for sentences I pointed out before, but your revision sentences still use *Since*.

Otherwise, the paper seems well written and my concerns have been met.

Decision letter (RSPB-2019-2269.R0)

07-Nov-2019

Dear Dr Christensen

I am pleased to inform you that your manuscript RSPB-2019-2269 entitled "Established rodent community delays recovery of dominant competitor following experimental disturbance" has been accepted for publication in Proceedings B.

The referee has recommended publication, but has also suggested some minor revisions to your manuscript. Therefore, I invite you to respond to their comments and revise your manuscript. Because the schedule for publication is very tight, it is a condition of publication that you submit the revised version of your manuscript within 7 days. If you do not think you will be able to meet this date please let us know.

- 1) A text file of the manuscript (doc, txt, rtf or tex), including the references, tables (including captions) and figure captions. Please remove any tracked changes from the text before submission. PDF files are not an accepted format for the "Main Document".

2) A separate electronic file of each figure (tiff, EPS or print-quality PDF preferred). The format should be produced directly from original creation package, or original software format. PowerPoint files are not accepted.

3) Electronic supplementary material: this should be contained in a separate file and where possible, all ESM should be combined into a single file. All supplementary materials accompanying an accepted article will be treated as in their final form. They will be published alongside the paper on the journal website and posted on the online figshare repository. Files on figshare will be made available approximately one week before the accompanying article so that the supplementary material can be attributed a unique DOI.

Sincerely,
 Professor Loeske Kruuk
 mailto: proceedingsb@royalsociety.org

Associate Editor
 Board Member
 Comments to Author:

There are some minor revisions suggested by the reviewer, and I am sure the copy editor will have more, but the major issues have been addressed. Even if there are some puzzles still left! - but you have discussed them, which is what the reviewers and I suggested.

Reviewer(s)' Comments to Author:

Referee: 2

Comments to the Author(s).
 Established rodent community delays recovery of dominant competitor following experimental disturbance by Christensen et al.

Proceedings of the Royal Society B: manuscript #RSPB-2019-2669

This is a revision of the paper I reviewed in the Spring. I did not have many serious criticisms of the original paper. I pointed out that the study needed to be repeated in time at least once to determine if the same delayed recovery of kangaroo rats on established rodent communities gives the same outcome as this one-time study and suggested at least pointing this out in the Discussion. The authors have done this. I also wanted to see some discussion of restoration ecology in the Discussion based on these findings, and this was done, although I think the topic could have been better and more fully explained. That said, what the authors have written is fine.

There still are minor edits to make:

Lines 211: Delete In order from the beginning of the sentence. It is clear if you write "To better..." Remove In order anywhere it occurs.

Lines 241, 304, etc.: Change since to because. Since is only used when referencing time periods, and because is used for causality. You did this for sentences I pointed out before, but your revision sentences still use Since.

Otherwise, the paper seems well written and my concerns have been met.

Author's Response to Decision Letter for (RSPB-2019-2269.R0)

See Appendix B.

Decision letter (RSPB-2019-2269.R1)

13-Nov-2019

Dear Dr Christensen

I am pleased to inform you that your manuscript entitled "Established rodent community delays recovery of dominant competitor following experimental disturbance" has been accepted for publication in Proceedings B.

Your article has been estimated as being 8 pages long. Our Production Office will be able to confirm the exact length at proof stage.

Open Access

You are invited to opt for Open Access, making your freely available to all as soon as it is ready for publication under a CCBY licence. Our article processing charge for Open Access is £1700. Corresponding authors from member institutions (<http://royalsocietypublishing.org/site/librarians/allmembers.xhtml>) receive a 25% discount to these charges. For more information please visit <http://royalsocietypublishing.org/open-access>.

Paper charges

Sincerely,
Proceedings B
<mailto:proceedingsb@royalsociety.org>

Appendix A

Response to referees

=====

Associate Editor

Comments to Author:

Any field study that has been going on as long as the Portal site study with mostly-consistent methods and plots, is valuable! I am glad that a new cohort of researchers has taken it over and is keeping it going. And the results you present are intriguing.

Response: Thank you!

However, the results are oddly incongruent with the focus of large parts of the Introduction and Conclusion. There is no evidence of alternative stable states, so why focus on this issue in the Introduction and Conclusion? I am just guessing - but perhaps you focused on alternative stable states in your NSF proposal? Whether or not this was the case, the Introduction and Conclusion of the paper should fit the actual results. Specifically, the Introduction, from line 77 through the following paragraph, is written as if you were providing the frame for a finding of alternative stable states. Lines 306 - 314 in the Conclusion are also about alternative stable states. I recommend rethinking and completely focusing the Introduction and Conclusion so they provide a frame for what you actually found: differences in the speed (and dynamics, see below) of recovery, with no evidence of a second stable state.

Response: Yes, admittedly the original intention of this project was to explore the possibility of “alternate stable states” in a rodent community. We agree that a re-framing was needed, and so we have re-written the Introduction and Conclusion sections without the vestigial references to alternate stable states.

You have some good text about the transient dynamics and the relative speed of recovery, which could make a better focus than alternative stable states. In line 326, you state that “Our results suggest not only that existing established species can alter or delay changes in biodiversity state, but also the reverse - that removal or disruption of an existing community [rodent removal treatment, I gather] may facilitate and/or speed up biodiversity shifts [kangaroo recovery, I gather].” I found this very interesting, and I think other readers would too. It might also provide a good bridge to discussing the restoration implications, which, as Reviewer 2 points out, you mention in the Introduction but do not come back to in the Discussion or Conclusion. Reviewer 2 considers that the difference in recovery times between the two treatments is actually too small to be important - this should be addressed if you discuss the results in the context of restoration.

Response: Thank you for your suggestions. We have re-written the Introduction and Conclusion to put this study into the context of transient dynamics. We also expanded on the implications of transient dynamics for restoration applications (lines 597-610), drawing from the literature on this topic. We agree with Reviewer 2 that the overall difference between treatments is small, and so we have tried to emphasize the fact that the presence of transient dynamics at all was surprising, and that transient dynamics provide opportunity for further unexpected variation and complex dynamics (lines 593-

595).

Reviewer 1 notes that in Fig 2B and 3B, almost all traces of the seasonal oscillations are gone. I concur with this reviewer's concern that information may be being lost due to smoothing. The rodent removal had much stronger oscillations than the other two (Fig 2A). And most of the time, there were more kangaroo rats in the rodent removal (blue) than in the kangaroo rat removal (gray) and sometime more than in the control (orange). You say (line 191 and ff) that kangaroo rat numbers in rodent removals (blue) converged with control numbers (orange) within 3 months - which they did, but then they bounced around, including an overshoot. Why did the rodent removal plots have so much greater oscillations in kangaroo rat numbers than the controls or rodent removals? Did this have anything to do with faster recovery there?

Response: We believe that the "overshoot" dynamics in rodent removals (blue) can be attributed to transient dynamics. Our additional analyses of population metrics also suggest that this overshoot was the result of higher immigration rates on those plots—possibly because there was not an established community to deter immigration. There is evidence in the literature (e.g. Neubert and Caswell 1997, and Caswell 2019, now included in the References section) that transient dynamics often include increased variation or initially moving away from the equilibrium point before approaching it. We have added a discussion of this phenomenon (lines 309-314).

As Reviewer 1 says, this paper leaves us with a puzzle: what caused the difference in the rate of kangaroo rat recovery between the two removal treatments? You eliminate differences in overall plant community composition and total resource availability. Reviewer 1 suggests some other possibilities. What do you think is the most likely explanation? I concur with this reviewer: "If they can provide more information, this should be an outstanding paper."

Response: We agree that this study leaves some puzzles unsolved. We have added new analyses (the population models presented in lines 291-300, and new figure 3) in line with some of Reviewer 1's suggestions in order to provide more context for the results we present. While we still don't have a "smoking gun" in terms of a mechanism which prevented kangaroo rats from recovering quickly on experimental treatment plots, these new analyses support the view that kangaroo rats had a behavioral aversion to the treatment plots, and did not disperse onto these plots at the same rate as onto the other plots. We propose in the manuscript that this aversion resulted from lower resource availability on those plots due to resources being sequestered by previous residents and therefore not available to immigrants, but we still cannot say for sure.

Referee: 1

The authors presented a multi-yr manipulated experiment on recovery of rodent communities. The results showed established species might impede the restoration of kangaroo rats. I appreciate the effort of such an experiment. However, the conclusion is not surprising. I would expect some results on the mechanism of slow recovery for the kangaroo removal treatment, but only the plant community was compared. For example, the demography, habitat use, behavior, dispersal process? Essentially, the authors did not answer why on earth the recovery or resembling-control-group is slower. Much more content is needed. If they can provide more information, this should be an outstanding paper.

1. The authors used count and metabolic flux to test restoration of kangaroo rats. The result is simple and straightforward, with many things unexplored. A deeper digging is thus necessary. Eg., for a community, the descriptions of the state are many, such as some common ones like diversity indices, etc. at population level, reproduction, age structure, sex ratio, etc. Pls do not misunderstand that I am asking for a full list of properties. I just suggest more exploration on what was exactly recovered.

Response: We performed several new analyses in order to provide deeper context for the results we initially presented. Specifically, we used the mark-recapture aspect of the study to perform multistrata population models (lines 291-300). The results of these models (summarized in the new Figure 3), especially the finding that kangaroo rats move onto the kangaroo rat+ treatment plots at a lower rate, support our hypothesis that kangaroo rats found these plots unappealing in some way. We regret that we do not have data on reproduction or age structure of the population, as this information may have helped further explain population dynamics on different treatments, but we feel that our additional models provide an additional step toward explaining the observed differences in population sizes on different treatments.

2. The control declined in the last two years. Any idea why? Except just saying a system phenomena.

Response: Unfortunately, we don't currently know what mechanism is behind the recent decline in kangaroo rat numbers on the control plots. Over the 40 years this study has been active, capture numbers have varied greatly in response to phenomena such as droughts and extreme flood events, and likely also due to unmeasured factors that are beyond the scope of the study such as disease and predation pressure. While it is unsatisfying not to have an answer for the decline on the control plots, it demonstrates the importance of having an appropriate control with which to compare treatments. By comparing the dynamics on treatment plots to dynamics on this time-varying control, we are confident that the effects we are describing can be attributed to the treatment alone.

3. I am not sure, but is there any possibility the rodents move between control and treatments. If yes, how does it affect the results?

Response: In our multistrata population models we included terms for animals moving between treatment types. Movement between treatments was only a significant factor for one species, *D. merriami*, and it was interesting that individuals appeared to preferentially move out of kangaroo rat+ plots and onto control or rodent+ plots. This

behavior supports our conclusion that kangaroo rats responded negatively to conditions on kangaroo rat+ treatment plots, and we have added a discussion of this to the results section (lines 364-376).

4. In fig. 2, the seasonal pattern is missing compared to non-kangaroo rodents in control group.is gam smooth not appropriate (need more knots in smooth function?) or actually missing?

Response: The ultimate upper limit of the “wiggleness” (complexity) of a smoother in a GAM fitted using penalized spline is the size of the basis expansion used, not the number of knots per se (the splines we used don’t have knots in the conventional sense). In our case the upper limit is related to the number of unique observation times in the data set. We used as large a basis as possible when estimating the reported models. To confirm whether the lack of seasonal variation might be due to the way we set up the GAMs, we tried several different parameterizations, where we decomposed time into explicit Year and Seasonal components to try to directly estimate the two separate signals. These models are more complex than the one we present in the paper as they involve two temporal smoothers (year and season) plus their smooth interaction, rather than a single smoother of Time. Model fits for these decomposed GAMs were highly uncertain and we encountered difficulties in estimating the models, suggesting that the model was over-parameterized given the available data. The model we present in the paper had better performance and fit based on several metrics, including AIC. The lack of seasonal signal in the control group in the model we present could reflect a real absence of this seasonal variation in abundance, or it may simply be that we lack the power to detect a seasonal signal in the control group, given the limited size of the available data set and the noise/sampling variability in the time series.

===

Referee: 2

Established rodent community delays recovery of dominant competitor following experimental disturbance by Christensen et al.

Proceedings of the Royal Society B: manuscript #RSPB-2019-0188

This study looked at the length of time for three species of kangaroo rats (*Dipodomys* spp.) to establish on 50 m² plots that either had an intact rodent community without kangaroo rats or on plots that had been kept free of all rodents. The authors used control plots with all species as a comparison. The work took place in desert habitat near Portal in southeastern Arizona. Multiple papers have been written from the experimental design at this site since the late 1970s. The basic outcome was that kangaroo rats established on plots without any rodents in about 3 mo, but it took about 2 y for a population of kangaroo rats to establish on plots with and otherwise intact rodent community.

The methodology is sound and the manuscript is well written. I have no major methodological or interpretive concerns. I do, however, have two minor comments and several small edits. Although this study has been internally replicated, this study needs to be repeated in time at least once to determine if the same delayed recovery of kangaroo rats on established rodent communities gives the same outcome as this one-time study. I am not suggesting that this study is fatally flawed, and although they allude to this problem in the Discussion, I think you need to more forcefully acknowledge that outcomes may be different at different times.

Response: Thank you. We agree that every experiment is context-dependent and there is increasing evidence that experiments in general can yield different results as conditions change, and our study has the same limitations in this regard as all experiments do. We have added a new final sentence to the conclusion (lines 611-615) that highlights the importance of repeating experiments like this one to assess how differences in resident community composition or other initial conditions could further impact observed transient dynamics.

Also, you point out that what you studied bears on restoration ecology, but you do not come back to this topic in the Discussion. I think this is an important topic, and I note that the time differences for kangaroo rats you found actually is unimportant to reestablishing *Dipodomys*. In fact, this is encouraging in that kangaroo rats will reestablish rather quickly even with multiple competitors at a site.

Response: Thank you for pointing out this limitation. We have added more discussion of the implications of our study for restoration/applied ecology (lines 598-611). You are correct that the “delay” in colonization that we observed was small in the grand scheme of this kangaroo rat population, but in this study we want to emphasize the fact that transient dynamics are possible (or even likely to occur) when populations are

manipulated for management purposes, and it's important to be aware of these surprises and provide enough time for them to play out.

Minor edits:

Lines 70, 99, 115, others: You use 'which' in these instances when the correct word is 'that.' That is used for restrictive clauses, and which is for unrestrictive clauses and is set off by a comma.

Line 110 and others: Remove the hyphen from behaviorally-dominant and other ly words. Words that end in ly are not hyphenated when paired with another adjective.

Line 126: 20-hectare change to 20-ha. Throughout the manuscript, use standard abbreviations for units.

Line 127: Capitalize desert in Chihuahuan Desert.

Line 133: Use only metric system throughout the paper.

Line 159: Change since to because. Since is only used when referencing time periods, and because is used for causality.

Throughout the manuscript, you use However to start sentences. Because however is a conjunctive adverb, it should lower case, and set off from the preceding sentence with a semicolon and followed by a comma (...; however,....).

Response: Thank you for pointing out these errors in grammar and syntax. They have been corrected.

Appendix B

Response to referees

Reviewer 2 pointed out some errors in grammar. These have been corrected. Attached is a tracked-changes version of the manuscript.